# Self-Assembly of Au-Fe_3_O_4_ Hybrid Nanoparticles Using a Sol–Gel Pechini Method

**DOI:** 10.3390/molecules26226943

**Published:** 2021-11-17

**Authors:** Jesus G. Ovejero, Miguel A. Garcia, Pilar Herrasti

**Affiliations:** 1Instituto de Magnetismo Aplicado, ‘Salvador Velayos’, UCM-CSIC-ADIF, Las Rozas, P.O. Box 155, 28230 Madrid, Spain; magarcia@icv.csic.es; 2Servicio de Dosimetría y Radioprotección, Hospital General Universitario Gregorio Marañón, 28007 Madrid, Spain; 3Instituto de Cerámica y Vidrio, ICV-CSIC, C/Kelsen 5, Cantoblanco, 28049 Madrid, Spain; 4Departamento de Química Física Aplicada, Facultad de Ciencias, Universidad Autónoma de Madrid, Cantoblanco s/n, 28049 Madrid, Spain

**Keywords:** Pechini method, click chemistry, self-assembly, nanoparticles, hybrid nanoparticles, iron oxide, gold nanorods, silica, magneto-plasmonic

## Abstract

The Pechini method has been used as a synthetic route for obtaining self-assembling magnetic and plasmonic nanoparticles in hybrid silica nanostructures. This manuscript evaluates the influence of shaking conditions, reaction time, and pH on the size and morphology of the nanostructures produced. The characterization of the nanomaterials was carried out by transmission electron microscopy (TEM) to evaluate the coating and size of the nanomaterials, Fourier-transform infrared spectroscopy (FT-IR) transmission spectra to evaluate the presence of the different coatings, and thermogravimetric analysis (TGA) curves to determine the amount of coating. The results obtained show that the best conditions to obtain core–satellite nanostructures with homogeneous silica shells and controlled sizes (<200 nm) include the use of slightly alkaline media, the ultrasound activation of silica condensation, and reaction times of around 2 h. These findings represent an important framework to establish a new general approach for the click chemistry assembling of inorganic nanostructures.

## 1. Introduction

Magneto-plasmonic nanoparticles (NPs) are a revolutionary family of nanostructures that combine phases with a ferromagnetic response and metallic phases with free electrons that can be optically excited as a surface plasmon resonance [1,2]. The possibilities that these nanostructures offer to the biomedical field in terms of dual contrast agents [3], theragnostic agents [4], or biosensing [5] are countless. However, single-phase NPs with good magnetic properties are generally bad conductors and vice versa [2]. For this reason, there is great urgency to develop new synthetic methods to create multicomponent NPs in which magnetic and plasmonic phases can be combined [6,7]. In this sense, the use of organic–inorganic interactions through click chemistry techniques appear to be the best solution to assemble inorganic phases with minimal disturbance of their individual properties [8].

One of the most intensively studied chemical routes for the synthesis of organic–inorganic hybrid nanoparticles is the sol–gel reaction [9,10]. This synthesis method controls the nucleation and growth of NPs by trapping the inorganic precursors in a homogeneous organic gel that regulates the formation steps and offers great versatility in size, shape, and crystallinity of the final product. However, the control of the kinetics and growth in this sort of synthesis requires a careful tuning of parameters, such as the solvents employed, temperature, concentration of precursors, pH, and agitation [11,12,13,14]. 

The development of sol–gel routes mediated by organic molecules have improved the quality and variability of the materials produced by this chemical route [15,16]. The Pechini method, for example [17], permits the creation of ceramic materials out of metals with no stable hydroxo species [18]. In this method, citric acid (CA) acts as a chelating agent for the metallic species, and ethylene glycol (EG) molecules create a soft lattice that connects the CA–metal complex via polyesterification [19,20]. The combination of both effects favors the distribution of doping metals in ceramics and improves the physicochemical properties in respect to other routes, such as the solid-state reaction process or the amorphous citrate process [21,22].

In 1997, Campero et al. used these two organic mediators for the first time in the growth of silica surfaces via the Stöber method [23]. They grew hybrid silica surfaces in which CA and EG acted as molecular connectors between silica precursors (tetraethyl orthosilicate (TEOS)), creating an organic–inorganic amorphous structure. Theoretical studies on the molecular dynamics showed that EG and CA connected hydrolyzed species of TEOS through ester bonds [24], as indicated in Figure 1.

This kind of specific interaction has been recently applied to the self-assembling of iron oxide nanoparticles (IONPs) coated with CA and Au nanorods (AuNRs) coated with thiolated polyethylene glycol (SH-PEG), respectively [25]. The hybridization of inorganic nanoparticles using silica matrices offers several advantages such as high versatility [26], the biocompatibility of the silica and the organic mediators employed [27,28,29], and the creation of a protective silica shell [30]. However, the mechanism of assembling and the chemical interaction between the coatings has not been studied in detail yet.

This paper analyzes the key parameters of the sol–gel hybridization reaction required to produce nanostructures with a limited size and appropriate colloid features. The transmission electron microscopy (TEM) and Dynamic Light Scattering (DLS) measurements showed the significant effects of the agitation method, the time of reaction, and pH in the final morphology of the nanostructures. Furthermore, the molecular mechanism of hybridization was analyzed using Fourier-transform infrared spectroscopy (FT-IR).

## 2. Results

### 2.1. Synthesis of Primary Nanoparticles and Surface Functionalization with PEG and CA

The Pechini sol–gel method studied in this article creates magneto-plasmonic hybrid nanostructures by combining IONPs and AuNRs prepared with tetrabutylammonium bromide (TBAB) and hexadecyltrimethylammonium bromide (CTAB) coatings, respectively. The NPs used as primary blocks for hybridization are presented in Figure 1. The CTAB used in the synthesis of AuNRs (Figure 1a) acts as an organic template that shapes the growth of the crystal facets to generate an average length of 39.7 nm (σ = 0.14) and an average diameter of 12.1 nm (σ = 0.21) that result in an aspect ratio of 3.4 (σ = 0.23). In the case of the IONPs (Figure 1b), the electrochemical settings chosen produced quasispherical nanoparticles with an average diameter of 14.9 nm (σ = 0.30). The shape of the AuNRs shifted the maximum optical absorption to the infrared region of the spectrum (725 nm), which is suitable for optical excitation in biological media. The sizes of the IONPs produced were ideal to maintain a high saturation magnetization (77.6 emu/g), preserving the superparamagnetic response of the sample. A more detailed characterization of these samples can be found in our previous works [31,32].

Before the hybridization process, the surface of each set of NPs was functionalized with the complementary molecules that mediate the Pechini reactions, such as CA and SH-PEG. Figure 2 shows the FT-IR and TGA curves of the AuNRs and IONPs before and after being coated. In the AuNR spectra (Figure 2a), the 719/730 cm^−1^ relates to CH_2_ rocking [33]; the 910/960 cm^−1^ peaks relate to the C-N group; and the three peaks 1461/1462/1485 cm^−1^ are produced by the scissoring of the CH_2_ and N-CH_3_ groups. The three groups of peaks disappeared after coating and new peaks associated to the skeletal vibration of the polymeric chain (1035 cm^−1^–1105 cm^−1^) appeared instead^34^. The additional peak observed at 950 cm^−1^ can be attributed to the stretching of the C-O-C groups of the PEG chains [34,35]. TGA curves presented in Figure 2c corroborated the functionalization of AuNRs. The CTAB coated AuNR sample (AuNR@CTAB) shows small percentages of weight loss at three different temperatures (180 °C, 265 °C, 460 °C), whereas the PEG coated AuNR (AuNR@PEG) suffers the main drop in weight at 365 °C. The amount of weight lost in the latter case is in agreement with the percentage reported in the bibliography for a single layer of PEG coating on AuNRs [35,36].

The IONP coatings were also analyzed by these techniques. In this case, the characteristic peaks of TBAB (2958 cm^−1^, 2875 cm^−1^, and 1474 cm^−1^) were substituted by 1715 cm^−1^ and 1618 cm^−1^ peaks associated with carboxylic groups of CA. Furthermore, the 1380 cm^−1^ observed before and after functionalization can be attributed to the bending deformation of CH_3_ groups in the TBAB coated IONP (IONP@TBAB) spectrum [37], or v_s_ (COO−) of the carboxylic groups in citric acid coated IONPs (IONP@CA). The resulting TGA curve was also modified after the functionalization of the IONPs. The IONP@TBAB presents a staircase loss of weight [38] that is substituted by a continuous drop centered at 240 °C in the case of the IONP@CA [39].

A complementary measurement to test the success of the functionalization is the *ƺ*-potential presented in Figure 3, which indicates a change in surface charge. At low pH, the *ƺ*-potential of the AuNR@CTAB is around +20mV and starts to decay for pH values over 9. After PEG coating, the surface charge becomes slightly negative and goes to neutral at pH 3. These values indicate that the colloidal stability of the AuNR@PEG is based on the steric repulsion of the polymeric chains of PEG instead of the electrostatic repulsion of the coatings. The IONPs also suffer a change in their surface charge after coating. The *ƺ* -potential of the IONP@TBAB varies from +30 mV at low pH to −30 mV at high pH, with an isoelectric point around pH = 6.5. The isoelectric point of the colloid shifts below 3 after coating IONPs with CA, reaching *ƺ* -potential values under -30mV for pHs over 4. The *ƺ* -potential measurements confirm the correct functionalization of the nanoparticles’ surfaces.

### 2.2. Effect of Key Hybridization Parameters on the Nanostructure Geometry

#### 2.2.1. Shaking Conditions

The first hybridization parameter studied was the shaking conditions applied during the reaction. Figure 4 displays the TEM images of hybrid nanostructures prepared under static conditions, magnetic stirring, and ultrasound mixing. The DLS measurements of the hydrodynamic size corresponding to each sample are presented below.

TEM images show that for static (Figure 4a) and magnetic stirring conditions (Figure 4b), the silica clusters contain several AuNR centers (dark contrast). DLS measurements confirm that both conditions create colloidal instable nanostructures with hydrodynamic sizes over 500 nm and polydispersity index (PDI) larger than 0.4. Only by carrying out the reaction inside an ultrasound bath was it possible to generate nanostructures with a single AuNR core surrounded by several IONPs. The hydrodynamic size registered by DLS (D_H_ = 266 nm, PDI = 0.20) suggests that the aggregates observed in TEM images are formed by smaller individual entities. 

The ultrasound bath disaggregated the NPs and prevented the sedimentation of the nanostructures. It has also been reported that the ultrasound waves promote the hydrolysis and condensation process of TEOS molecules [40], avoiding the cross-linking between NPs [41,42] and enhancing the stability of the silica coatings [42].

#### 2.2.2. Reaction Time

In order to analyze the different steps of the hybrid nanostructure formation, the hybridization was stopped at different times by diluting the reaction with an excess of ethanol and instantly washing by centrifugation. Figure 5 shows the hybrid nanostructures obtained for reaction times of 30, 60, and 180 min. At 30 min, the silica forms a thin layer of 2.9 ± 0.3 nm around the AuNR, in which several IONPs appear to be inserted. The TEM images showed that the external surface of the IONP does not present signs of silica formation. At 60 min, most of the AuNR and IONP appears completely surrounded by a common silica shell, and at 120 min the hybridization is completed.

According to DLS measurements, the D_H_ of the nanostructures does not vary significantly for different reaction times, which suggests that the aggregates observed in TEM images are not a consequence of silica cross-linking between NPs. Instead, they seem to be formed by magnetic aggregation and the coffee-stain effect during TEM grid preparation [43].

The thickness of the silica shell observed at 30 min (~4 nm) matches with the size of the single layer of 6 kDa PEG coating estimated from the TGA curves of AuNRs in Figure 2 [44]. Previous studies have observed that the hydrogen bonds formed between PEG and TEOS molecules promote the condensation of TEOS molecules and can be used to control the size of silica coatings on NPs [45]. The presence of IONPs in the silica shell around AuNRs at the earliest stages suggests that self-assembly of both components is driven by the cross-bonding between PEG and CA coatings instead of the fusion of IONP and AuNR silica shells.

#### 2.2.3. pH Conditions

The third parameter considered in this study was the effect of the pH. In this case, different amounts of NH_3_ were added to the solution to establish different basicities in the medium since the precise pH could not be determined due to the mixture of solvents (ethanol/water).

The nanostructures presented in Figure 6a show that the silica coating results were highly inhomogeneous in the absence of ammonium. The silica condensated under these conditions, leaving some parts of the AuNRs surface exposed and trapping several Au cores in a common silica matrix. DLS measurements reflect such inhomogeneity (PDI = 0.82) with peaks at a few nanometers, likely associated with the Brownian rotation of unlinked AuNRs [46], and the peaks of several microns generated by microparticles, as displayed in Figure 6a. 

On the other extreme of the pH scale, an excessive amount of ammonium generates large matrices of silica in which multiple AuNRs and IONPs are imbedded. The sizes of these nanostructures are over 500 nm, although they maintain a relatively low dispersity (PDI = 0.14). Hybrid nanostructures with a core–satellite arrangement and hydrodynamic diameters bellow 300 nm were only obtained using the appropriate amount of ammonium. As indicated in previous sections, the PDI is approximately 0.1 in these kinds of preparations.

The dependence of the hybridization process with the pH can be explained by taking into account the sol–gel reactions involved in the Stöber method. It is generally accepted that increasing the pH of the reaction helps to partially deprotonate the silanol groups of TEOS and promotes their hydrolysis [47]. If condensation takes place between two fully hydrolyzed TEOS molecules, the resulting product (OH)_3_ Si–O–Si(OH)_3_ has six sites for further condensation [10], which promotes the branched growth of silica structures. On the contrary, under acidic conditions, the silica growth tends to form linear bonds between TEOS centers that create fibrillar structures of silica [48]. It must also be taken into account that at high pH, the electrostatic repulsion of the silica particles is stronger and certain aggregation effects can be avoided [48].

According to these results, our hypothesis is that at low pH, the silica condensation produces a crosslinked growth that connects several NPs and cancels the self-assembling effects mediated by PEG and CA. However, under basic conditions, the condensation of SiO_2_ is too fast to be controlled by organic mediators and the silica matrix grows randomly, embedding several AuNRs and cancelling the self-assembly of the NPs. Thus, only the proper amount of base allows the condensation to slow down so that the mediators can control the assembling process and avoid the fibrillar growth of silica coatings.

## 3. Molecular Bonding Mechanism for the Best Hybridization Conditions

The FT-IR spectrum of the hybrid nanostructures, prepared under optimum conditions (Figure 7), shows that the most important peaks of PEG and CA molecules observed in Figure 2 remain after the hybridization. The main differences observed are these:
(i)The broad band at 3450 cm^−1^ grows in intensity in respect of the IONP@CA pattern due to the contribution of -OH groups of PEG and hydrolyzed TEOS.(ii)The contribution of stretching the vibrational mode of Si-OH shifts the small absorption peak observed at 950 cm^−1^ in the AuNR@PEG pattern to 961 cm^−1^.(iii)The low intensity shoulder observed at 1729 cm^−1^, which is associated to the ester bond between CA and PEG.(iv)The two differentiated peaks at 1105 cm^−1^ and 1035 cm^−1^ observed in the AuNR@PEG pattern become a single broad peak centered at 1105 cm^−1^, with two shoulders at 1179 cm^−1^ and 1221 cm^−1^ (deconvolution in SI).

The last effect is likely a consequence of the absorption of the longitudinal asymmetric stretching (1180 cm^−1^) and the transverse vibration (1075 cm^−1^) [49] of the Si-O-Si groups generated during the growth of the silica shell. However, due to the multiple bands overlapping, it is not possible to resolve the presence of the 1100 cm^−1^ peak associated with the asymmetric stretching of the Si-O-C bond [50,51].

To the best of our knowledge, two conjugation mechanisms have been proposed so far for the hybrid silica structures based on TEOS, CA, and PEG [23,24,52,53,54]. On one hand, Cardoso et al. proposed a transesterification through the TEOS molecule, such as the one displayed in Figure 1c. On the other hand, Zaharieva et al. [54] indicated that depending on the molar amount of TEOS:EG:CA, the transesterification can take place between CA and EG without the mediation of TEOS molecules. The characteristic FT-IR peak for the CA-PEG ester bond appears at 1730–1735 cm^−1^ and 1196–1205 cm^−1^. The low intensity of the peak observed at 1729 cm^−1^, and the peak at 1221 cm^−1^ deconvoluted from the FT-IR pattern of hybrid nanostructures, suggests that Cardoso’s mechanism is the main hybridization process in this case. 

## 4. Materials and Methods

### 4.1. Reactants

Ethanol, hexadecyltrimethylammonium bromide (CTAB), chloroauric acid (HAuCl_4_), sodium borohydride (NaBH_4_), silver nitrate (AgNO_3_), sodium hydroxide (NaOH), ascorbic acid, citric acid (CA), poly(ethylene glycol) thiol (PEG-SH), and tetraethyl orthosilicate (TEOS) were purchased from Sigma-Aldrich (www.sigmaaldrich.com, accessed on 01/01/2021, San Luis, MI, USA). Ultrapure water, tetrabutylammonium bromide (TBAB), and ammonia (28%, NH_3_) were purchased from PanReac AppliChem (www.itwreagents.com, accessed on 01/01/2021, Barcelona, Spain). Fe foils were purchased from Goodfellow (www.goodfellow.com, accessed on 01/01/2021, Goodfellow, Huntingdon, UK). Citric acid was purchased from Merk (Darmstadt, Germany).

### 4.2. Synthesis of Au NR

AuNRs were prepared according to the El-Sayed method [55]. Briefly, Au seeds were prepared by gently mixing 5 mL of CTAB 0.1 M and 0.25 mL of HAuCl_4_ 10 mM. Au ions were reduced by adding 0.3 mL of 10 mM ice cooled NaBH_4_ drop by drop, and the solution was left for 1 h in a thermostatic bath at 30 °C to complete the seed growth.

The growth solution was prepared by mixing 250 mL of CTAB 0.1 M with 12.45 mL of HAuCl_4_ (10 mM), and 2 mL of AgNO_3_ (10 mM). One minute before adding Au seeds, 1.2 mL of ascorbic acid 0.1 M was added to the solution. The growth of the AuNRs was initiated by adding 60 µL of seed and the mixture was sat overnight at 30 °C. Once grown, the suspension of AuNRs was cleaned by triple centrifugation of 10000 rpm for 30 min. All solutions described were prepared in ultrapure water.

The coating of AuNRs with SH-PEG was performed by adding 23.2 mg of SH-PEG (Mw = 6000) to 8.75 mL of a concentrated colloid of AuNR (OD_400nm_ = 2, (Au) = 0.83 mM)). SH-PEG was previously dissolved in 2.5 mL of water with the help of an ultrasound (US) bath for 25 min. The mixture was stirred for 6 h to complete the coating. After the pegylation, the samples were centrifugated at 10,000 rpm for 30 min and the medium was progressively substituted with pure ethanol.

### 4.3. Electrochemical Synthesis of IONP

IONPs were prepared by means of two flat iron electrodes submerged in a conductive solution of tetrabutylammonium bromide at 40 mM. A current of 45 mA cm^−2^ was applied between the two electrodes for 30 min, keeping the solution thermostated at 5 °C. One of the electrodes (2 cm^2^) acts as an anode producing the oxidation of iron–iron ions, whilst the other acts as a cathode (8 cm^2^), causing the reduction of water for the formation of the OH- group [31]. The iron electrodes were faced with a 1 cm gap between them. The solution was vigorously mixed using a magnetic stirrer to promote the precursor diffusion.

After three magnetic decantations and cleaning with distilled water (18.2 MΩ cm^−1^), the IONPs were immediately coated with citric acid by adding 25 mL of 0.1 M citric acid to the NPs whilst stirring. After adjusting the pH to 5, using diluted NaOH, the dispersion was heated up to 80 °C for 2 h. IONPs were cleaned again by triple magnetic decantation.

### 4.4. Synthesis of Hybrid AuNR-IONP

Before hybridization, the concentration of AuNRs in ethanol was adjusted to 0.34 g_Au_ L^−1^ of Au (OD_400nm_ = 2.4) and IONPs to 0.7 g_Fe_ L^−1^. The Fe concentration in the IONP colloid was estimated by inductively coupled plasma optical emission spectroscopy (ICP-OES).

Hybrid AuNR-IONP were prepared in a 12.5 mL plastic round-bottom tube, adding 0.286 mL of IONP dispersion to 1 mL of AuNR to establish a solvent ratio of ethanol/water ~3.5. The mixture was manually shaken for few seconds and sonicated for 5 min. Then, 0.286 mL of TEOS was added and the dispersion was shaken and sonicated for another 5 min. The standard hybrid NPs were prepared by raising the pH of the sample with 5 μL of NH_3_ (28%) and introducing the tube to an ultrasound bath for 120 min whilst maintaining the temperature bellow 50 °C. The parameters studied varied in the following ways:The samples analyzed in the study of the shaking conditions were prepared by sitting the plastic tubes, stirring the solution with a magnetic stirrer, and introducing the tube to an ultrasound bath. The rest of parameters were maintained as the standard.The samples which analyzed the time of reaction were left in the ultrasound bath for 30, 60 and, 120 min. The rest of the parameters were maintained as the standard.The samples analyzed in the study of the pH were prepared using 2.5, 5, and 10 μL of NH_3_. The rest of parameters were maintained as the standard.The resultant hybrid NPs were cleaned by triple centrifugation for 30 min at 3000 rpm. Supernatant was discarded and the hybrid NPs were stored in pure ethanol for the analysis.

### 4.5. Characterization

The size and geometry of hybrid NPs were studied using a transmission electron microscopy (TEM) JEM-2100F (JEOL Ltd.). The average size of each collection of NPs was estimated by measuring more than 100 NPs and fitting the data with the log-normal distribution presented in Equation (1).
(1)$$f\left(x\right)=\frac{1}{x\sqrt{2\pi \sigma^{2}}}e^{-\frac{{\left(\ln\left(x\right)-\langle x_{0}\rangle \right)}^{2}}{\sqrt{2\sigma^{2}}}}$$
where <*x*_0_> is the center value of the distribution and *σ* is the distribution width. 

Fourier transformed infrared spectra (FT-IR) were measured in a Bruker IFS 66V-S using a frequency range of 4000–500 cm^−1^. The colloids were lyophilized, mixed with KBr dry powders, and compressed into a circular disk to prepare the samples for FT-IR measurements. Thermogravimetric analysis (TGA) was obtained in a DSC/DTA/TGA Q600 module from TA Instruments with a heating rate of 10 °C/min and N_2_ atmosphere.

Colloidal features such as hydrodynamic diameter (D_H_) and polydispersity index (PDI) were obtained by means of Dynamic Light Scattering (DLS) a Zetasizer Nano (Malvern) using the values of scattered light intensity (intensity mode). The *ƺ*-potential measurements were performed by dispersing the colloids in a KNO_3_ solution of 10^−2^ M and adjusting pH values with HNO_3_ and KOH.

## 5. Conclusions

The Pechini sol–gel synthesis of hybrid nanoparticles based on the interaction of CA, PEG and, TEOS requires the careful control of synthesis parameters, such as shaking conditions, reaction time, or pH. Our findings show that PEG and CA interactions through silicon centers of TEOS are responsible for the self-assembly process. The best conditions found for the synthesis of AuNR@IONP@SiO_2_ nanostructures consist of carrying out the reaction in an ultrasound bath for 120 min using ~0.1 volume percentage of ammonium. Using these conditions, the colloid of hybrid nanostructures produced present a core–satellite structure covered by a silica shell, a controlled hydrodynamic between 250 and 300 nm, and PDIs smaller than 0.20. Thus, we propose a modified Pechini method as a versatile and innovative route for the controlled self-assembly of hybrid nanostructures.

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
