# Peer review of "Self-Assembly of Au-Fe_3_O_4_ Hybrid Nanoparticles Using a Sol–Gel Pechini Method"

_molecules, 2021, doi:10.3390/molecules26226943_

Round 1

Reviewer 1 Report

  1. The abstract is a narrative of rationale and motivation for the study without any information about the result of the study. The authors should present the results obtained from the study in the abstract.
  2. Only three keywords were cited but two of them are generalized keywords without being specific to the manuscript.
  3. Page 2, line 35, change “properties8” to “properties [8]”.
  4. Pages 2-3, lines 71-83. The discussion of the TEM images on page 3, Figure 1 focussed only on their average diameter but there is no mention of the particle shapes even though the two images have different shapes. The authors should discuss the particle shapes and reasons while different synthetic methods leads to the formation of particles with different shapes.
  5. Page 3, line 92, change “the triplet” to “the three peaks’.
  6. Page 3, lines 97-101: The discussion of the TGA does not add any value to the paper and should be deleted because the authors could not calculate the percentage degradation of each organic moiety.
  7. Page 5, line 148, the title of Figure 4 is incorrect. It is TEM images and zeta potential showing the dependence of particle sizes on hybridization potential.
  8. Particles sizes in the TEM images presented in Figures 4, 5 and 6 should be measured and the images should be discussed via-a-vis the particle sizes with respect to the hybridization parameters, reaction time and changes in pH. What are the effects of change in hybridization parameters, reaction time and pH on the particles size and crystallinity?
  9. The author should carry out powder X-ray diffraction patterns of the particles presented in Fig. 1A&B, 4A, B &C, 5A, B &C, and 6A, B &C. Without the XRD, it is practically impossible to know the exact nature of the “Au-Fe3O4 hybrid nanoparticles”. Hence any other analysis on them is not useful scientifically.
  10. Page 10, conclusions. There is no conclusion in this manuscript but merely a description of what the author did not the results obtained from the study.

Reviewer 2 Report

I carefully read the manuscript "Self-assembly of Au-Fe3O4 hybrid nanoparticles using a sol-gel Pechini method". The manuscript structure is concise; I would suggest widening the explanation of why this work (and the modulation of the studied parameters) can help optimize such hybrid performances. For this flaw and other criticisms (listed below), I recommend the publication after major revisions.

  • What does OD400 nm stand for?
  • In Scheme 1, shouldn't the polymerization involve the presence of another atom of Si, connected to another oxygen, linking the CA moieties?
  • Regarding Materials and Methods section: Although the template of MDPI articles generally reports this section at the end of the "Results" section, I suggest including some general information about the synthesis procedure before the results. Otherwise, it isn't easy to understand the outcomes related to the organic parts attached to nanoparticles and silica (i.e., N-containing groups in FTIR spectra).
  • Is the storage of the hybrid in ethanol due to the non-stability/aggregation trend of the particles? Could this fact represent a problem for applications?
  • In the "Results" section, the sentence "The magnetic and plasmonic characterization of these samples can be found in our previous works" should be slightly explained by a summary of previous results in order to be immediately clearer for the reader.
  • The numerical data within TGA graphs (Fig.2 c and d) are very small, they should be modified to be clearer.
  • Regarding FTIR, why was not the peak at 1715 cm-1 attributed to carboxylic groups of CA?
  • Could the author clarify the sentence "This values indicate that the colloidal stability of AuNR@PEG is based on the steric repulsion of the polymeric chains of PEG instead of the electrostatic repulsion of the coatings". Why?
  • Fig 4: How did the author distinguish the silica cluster from IONP?
  • I would add a specific scheme reporting all the moieties involved in this Au-Fe hybrid, since a general scheme is reported (Scheme1), and the reader should imagine how all these organic and inorganic components are assembled and eventually interact.
  • Why is the characterization in section 3 isolated from the others? Moreover, the shifted peak is indicated at 961 cm-1 in the spectrum and at 967 cm-1 in the text, why? And why does the peak is shifted?
  • Figure s1 of SI is not presented anywhere in the main text.

Round 2

Reviewer 2 Report

The manuscript has been improved and the authors reply to all comments. 
